# Physical Performance and Sports Genetics: A Systematic Review of Candidate Gene Polymorphisms Involved in Team Sports

**DOI:** 10.3390/genes16091079

**Published:** 2025-09-15

**Authors:** Raluca Mijaica, Dragoș Ioan Tohănean, Dan Iulian Alexe, Lorand Balint

**Affiliations:** 1Department of Physical Education and Special Motricity, Faculty of Physical Education and Mountain Sports, Transilvania University of Brașov, 500068 Brașov, Romania; raluca_mijaica@unitbv.ro (R.M.); lbalint@unitbv.ro (L.B.); 2Department of Motric Performance, Faculty of Physical Education and Mountain Sports, Transilvania University of Brașov, 600115 Brașov, Romania; 3Department of Physical and Occupational Therapy, Faculty of Movement, Sports and Health, Sciences, Vasile Alecsandri University of Bacău, 600115 Bacău, Romania; alexedaniulian@ub.ro

**Keywords:** candidate gene, genetic polymorphism, team sports, athletic performance, injury risk, genotype score, personalized training, sport selection

## Abstract

**Background/Objectives**: This systematic review aimed to gather the most recent evidence regarding the link between genetic polymorphisms and physical performance in team sports, with a focus on the practical utility of this information for athlete selection, training personalization, and injury prevention. **Methods**: Sixteen studies published between 2018 and 2025 were analyzed and selected from six international databases, in accordance with the PRISMA guideline. Only English-language studies were included, which evaluated active athletes in team sports and investigated associations between genetic variations, such as Actinin Alpha 3 (*ACTN3 R577X*), Angiotensin I Converting Enzyme (*ACE I/D*), Peroxisome Proliferator-Activated Receptor Alpha (*PPARA*), Interleukin 6 (*IL6*), and Nitric Oxide Synthase 3 (*NOS3*), and physical performance parameters. The methodological quality of the studies was assessed using the Q-Genie tool, with all studies scoring over 45 across all 11 items, indicating high quality. **Results**: The *ACTN3* and *ACE* genes stood out due to their consistent association with traits such as strength, speed, endurance, and recovery capacity. Other genes, such as *PPARA*, Fatty Acid Amide Hydrolase (*FAAH*), Angiotensinogen (*AGT*), and *NOS3*, complemented this genetic profile by being involved in the regulation of energy metabolism and injury predisposition. An increasing number of studies have begun to adopt cumulative genotype scores, suggesting a shift from a monogenic approach to complex predictive models. **Conclusions**: The integration of genetic profiling into the evaluation and management of athletes in team sports is becoming increasingly relevant. Although current evidence supports the applicability of these markers, robust future research conducted under standardized conditions is necessary to validate their use in sports practice and to ensure sound ethical standards.

## 1. Introduction

The ongoing process of optimizing performance in team sports has led to an increasingly prominent trend toward understanding and utilizing knowledge about the biological mechanisms that support physical abilities. Even though traditional factors such as training, nutrition, and psychological aspects remain the main pillars for achieving notable athletic results, current advances in genomics have highlighted the existence of a relevant genetic component involved in the manifestation of interindividual differences in motor performance [1]. Thus, it can be said that physical performance in any sports discipline, including team sports, is influenced by a complex combination of genetic and environmental factors. Over the past 20 years, sports genetics has focused significantly on identifying genetic polymorphisms. These can provide physiological advantages in sports activities, with genetic markers being involved in mechanisms such as energy metabolism, muscle contractility, cardiovascular adaptation, and post-exercise recovery [2].

As previously mentioned, over the past two decades, research has focused on identifying candidate genes involved in the regulation of key physiological processes such as muscle strength, oxygen utilization, inflammatory response, and neuroplasticity [3]. Genes such as *ACTN3*, *ACE*, Peroxisome Proliferator-Activated Receptor Gamma Coactivator 1-Alpha (*PPARGC1A*), and *NOS3* are among the most studied regarding their role in sports performance. Thus, sprinting ability, muscle strength, and aerobic endurance, understood as highly relevant factors for physical and athletic performance in team sports, have been correlated with polymorphisms in the *ACTN3* and *ACE* genes [4]. In this context, the *ACTN3* gene is typically associated with strength and speed, while the *ACE* gene, through the I allele, is believed to be involved in blood pressure regulation and endurance, being correlated with better metabolic efficiency and performance in endurance sports. The *R577X* polymorphism in *ACTN3* influences muscle fiber composition and has been linked to performance in elite athletes, particularly those engaged in power sports [5]. The same *R577X* polymorphism in the *ACTN3* gene determines the expression of the α-actinin-3 protein, which is present in type II (fast-twitch) muscle fibers. Athletes with the RR genotype tend to have a greater predisposition for power and speed sports, whereas those with the XX genotype may have an advantage in endurance activities [6]. Another major polymorphism studied is *ACE I/D*, where the I allele is frequently associated with superior performance in endurance sports, and the D allele is correlated with anaerobic and strength capacities [5]. In a team context, these genetic variations can influence player positioning and personalized training strategies. Furthermore, genes such as *PPARGC1A*, involved in mitochondrial biogenesis and energy metabolism, and *IL6*, linked to inflammatory response and recovery, offer an integrated perspective on how genetics can impact both performance and injury risk [7,8].

Although numerous studies in the scientific literature refer to the link between genetics and athletic performance, systematic research specifically analyzing team sports remains relatively scarce. Team sports such as football, rugby, handball, and hockey are characterized by a high level of functional demand and an increased influence of situational factors, where polygenic profiles may influence adaptive responses to training stimuli, stress conditions in decision-making, and injury risk [9]. Varillas-Delgado et al. (2022) argue that the determinism between genes and performance in team sports can significantly contribute to improving selection strategies, training personalization, and injury prevention [4]. Moreover, the integration of genetic screening tools into sports monitoring systems has generated debates concerning the ethical and practical implications of using Deoxyribonucleic Acid (DNA) to guide the development of athletic talent [4]. Another interesting and specific aspect in the context of high-effort, contact-based team sports in association with genetics refers to certain athletes’ predisposition to injuries [10]. For instance, Ryan-Moore, Mavrommatis, and Waldron (2020) pointed out that certain types of candidate genes are linked to fracture risk and tissue resilience, suggesting that genetic predisposition may modulate biomechanical vulnerability under intense effort and elite-level performance conditions [10]. These genetic variations become particularly relevant in the context of team sports, where physiological demands are diverse and dynamic. They can be used to better understand individual responses to training, tolerance to intermittent effort, and the capacity to perform repeated high-intensity actions. A number of recent studies suggest that integrating genetic profiles into the selection and development of athletes may contribute to optimizing performance at both the individual and team levels [11,12]. However, the research indicates that genetic markers, when integrated with discernment into the study of relevant athletic outcomes, may have complementary predictive value, without serving as decisive factors in determining performance [13,14].

In team sports, performance depends not only on athletes’ individual speed or endurance abilities but also on cognitive, motivational, and neuromuscular coordination factors. Therefore, analyzing genetic factors in this context can contribute to training personalization, injury prevention, and the selection of athletes for specific positions on the field [7]. In particular, team sports provide an ideal framework for understanding gene–environment interactions, as physical and tactical demands vary significantly depending on the sport, position, and playing style. Thus, integrating genetic profiles into performance analysis can offer a holistic and evidence-based approach to optimizing athletic potential.

As interest in sports genetics has grown, a significant body of research has emerged concerning the role of genetic polymorphisms in athletic performance [13,14,15]. However, studies are often heterogeneous in terms of the populations analyzed, methods used, and sports investigated. In team sports, this diversity is further amplified by the complexity of physical activity and the influence of other factors such as tactics, coordination, and psychosocial interactions. In this context, the need becomes increasingly evident for analyses focused on a comprehensive understanding of these specific relationships involving relevant genetic associations for team sports, where types of effort and physiological demands are varied and interdependent [4,7,10].

Thus, the present study aims to conduct a systematic review of polymorphisms in candidate genes associated with physical performance in team sports, highlighting functional associations, predictive aspects, and future directions in genetic analysis applied to the field of elite sports.

## 2. Materials and Methods

### 2.1. Experimental Approach to the Problem

This systematic review was designed to synthesize evidence on the associations between candidate gene polymorphisms and physical performance in team sports. The process was conducted in accordance with the PRISMA guidelines (Preferred Reporting Items for Systematic Reviews and Meta-Analyses) [16,17] and followed the established recommendations for conducting systematic reviews in sports science [18], aiming for a rigorous, transparent, and reproducible analysis of the existing literature. The main objective was to critically evaluate studies that investigated correlations between genetic variations—particularly in the *ACTN3*, *ACE*, *PPARGC1A*, and *NOS3* genes—and performance parameters in the context of team sports. The underlying hypothesis was that certain polymorphisms may influence performance through physiological mechanisms such as strength, endurance, metabolic adaptation, or recovery.

The methodological process included formulating the research question (which genetic polymorphisms of candidate genes are associated with physical performance in team sports, and what is their functional relevance in a sports context?), establishing selection criteria, conducting a systematic search in scientific databases, assessing the quality of studies, and extracting relevant data. Through this approach, this study provides an integrative view of genetic influence in team sports and identifies useful directions for future research.

### 2.2. Information Sources

The literature search process was conducted between December 2024 and May 2025, following the PRISMA methodological framework for systematic reviews, and included the majority of studies published between 1 January 2018 and 29 May 2025. The last access to the databases was performed on 30 May 2025. Eligible articles were identified in the following international databases: Web of Science, Scopus, MEDLINE/PubMed, SPORTDiscuss, ProQuest Central, and Google Scholar, with only English-language studies being included. Additionally, to identify further studies, the reference lists of relevant articles were manually checked.

### 2.3. Search Strategy

Based on the PICO framework (Patient, Problem, or Population—Intervention or Exposure—Comparison, Control, or Comparator—Outcome[s]) [19], a search strategy was developed to ensure the identification of studies and a systematic coverage of the relevant literature (Table 1).

Study selection and evaluation were conducted independently by two researchers. In the interest of transparency, the authors were not blinded to journal names, authors, or study results. The keywords used in our search included combinations of the following thematic indicators: (“genetic polymorphism” or “gene variant” or “genetic markers” or “*ACTN3*” or “*ACE*” or “*PPARGC1A*” or “*NOS3*”) and (“team sports” or “football” or “rugby” or “handball” or “hockey” or “basketball” or “volleyball”) and (“performance” or “endurance” or “strength” or “power” or “aerobic capacity” or “injury risk” or “sports performance prediction”).

Across all consulted databases (Web of Science, Scopus, MEDLINE/PubMed, SPORTDiscuss, ProQuest Central, and Google Scholar), the following filters were applied: English language, original studies published in scientific journals, active athletes in team sports, and publication period between 2018 and 2025. Searches were conducted within the title, abstract, and keywords fields, where this option was available in the respective databases. For Google Scholar and ProQuest Central, the queries were applied to the full text. As previously mentioned, to maximize the relevance of the results, the reference lists of the identified articles were manually screened to identify additional eligible sources.

The search strategy was adapted for each database using the following queries:PubMed/MEDLINE: (“genetic polymorphism” [Title/Abstract] or “gene variant” [Title/Abstract] or “genetic markers” [Title/Abstract] or “*ACTN3*” [Title/Abstract] or “*ACE*” [Title/Abstract] or “*PPARGC1A*” [Title/Abstract] or “*NOS3*” [Title/Abstract]) and (“team sports” [Title/Abstract] or “football” [Title/Abstract] or “rugby” [Title/Abstract] or “handball” [Title/Abstract] or “hockey” [Title/Abstract] or “basketball” [Title/Abstract] or “volleyball” [Title/Abstract]) and (“performance” [Title/Abstract] or “endurance” [Title/Abstract] or “strength” [Title/Abstract] or “power” [Title/Abstract] or “aerobic capacity” [Title/Abstract] or “injury risk” [Title/Abstract] or “sports performance prediction” [Title/Abstract]).Scopus: TITLE-ABS-KEY (“genetic polymorphism” or “gene variant” or “genetic markers” or “*ACTN3*” or “*ACE*” or “*PPARGC1A*” or “*NOS3*”) and TITLE-ABS-KEY (“team sports” or “football” or “rugby” or “handball” or “hockey” or “basketball” or “volleyball”) and TITLE-ABS-KEY (“performance” or “endurance” or “strength” OR “power” or “aerobic capacity” or “injury risk” or “sports performance prediction”).Web of Science: TS = (“genetic polymorphism” or “gene variant” OR “genetic markers” or “*ACTN3*” or “*ACE*” or “*PPARGC1A*” or “*NOS3*”) and TS = (“team sports” or “football” or “rugby” or “handball” or “hockey” or “basketball” or “volleyball”) AND TS = (“performance” or “endurance” or “strength” or “power” or “aerobic capacity” or “injury risk” or “sports performance prediction”).SPORTDiscus: AB (“genetic polymorphism” or “gene variant” or “genetic markers” or “*ACTN3*” or “*ACE*” or “*PPARGC1A*” or “*NOS3*”) and AB (“team sports” or “football” or “rugby” or “handball” or “hockey” or “basketball” or “volleyball”) and AB (“performance” or “endurance” or “strength” or “power” or “aerobic capacity” or “injury risk” or “sports performance prediction”).ProQuest Central and Google Scholar: the query was applied to the full text, as advanced filtering options were limited, using the same combination of terms.

The search returned a total of 355 articles (343 from automated searches and 12 from the reference lists of already identified articles). The distribution across databases was as follows: PubMed/MEDLINE—61 articles, Scopus—62 articles, Web of Science—45 articles, SPORTDiscus—28 articles, ProQuest Central—71 articles, and Google Scholar—76 articles. After the removal of 40 duplicates, 315 articles remained for title and abstract screening. In addition, 12 relevant publications were identified and included from the reference lists of eligible articles.

### 2.4. Eligibility Criteria

The selection of articles was based on their relevance to the proposed topic, and in order to ensure the quality and relevance of those included in this systematic review, clear inclusion and exclusion criteria were defined.

#### 2.4.1. Inclusion Criteria

Scientific articles were included if they met the following conditions:Original studies, published in English only;Studies investigating associations between genetic polymorphisms and physical performance in team sports disciplines such as football, handball, rugby, hockey, volleyball, and basketball;Studies containing detailed data on specific candidate genes (e.g., *ACTN3*, *ACE*, and *AGT*);Studies involving participants from athletic samples such as juniors, professional athletes, or elite-level competitors actively involved in organized competitions;Studies reporting data on at least one physiological performance parameter (e.g., strength, speed, endurance, recovery time, aerobic/anaerobic capacity, and injury risk).Full-text availability of the articles, either through open access or institutional access (e.g., via university subscriptions).

#### 2.4.2. Exclusion Criteria

To clearly delimit the variables analyzed, the following types of studies were excluded from the final selection:Studies focused exclusively on individual sports or those addressing general topics in physical education without a clear link to performance in team sports;Non-scientific articles, such as personal opinions, editorials, letters to the editor, or short communications that do not provide rigorous and verifiable empirical data;Incomplete studies or those without full-text access, as well as studies that mentioned the genetic markers analyzed but provided data that was too limited or insufficient for comparative evaluation;Articles published in languages other than English.

### 2.5. Study Selection and Data Extraction Process

The study selection process was structured into two main phases, in accordance with the PRISMA guidelines and best practices in systematic research. All scientific papers identified through the search were reviewed and selected using a two-step structured approach to ensure the accuracy and relevance of the studies included in this scientific endeavor. In the first phase, the titles and abstracts of all articles identified through the database search were screened to eliminate irrelevant papers, duplicates, or studies that did not meet the basic criteria. In the second phase, potentially eligible articles were reviewed in full to clearly determine whether they met the previously established eligibility criteria, both from a methodological standpoint and in terms of scientific relevance to this study’s objective.

After reviewing all databases (Web of Science, Scopus, MEDLINE/PubMed, SPORTDiscuss, ProQuest Central, and Google Scholar), a total of 355 articles (343 + 12) were screened. In the first step, 40 duplicate articles were identified and removed. After this step, the authors assessed whether each of the remaining 315 articles met all inclusion criteria, which led to the exclusion of 256 articles. Full-text evaluation was then performed, leading to the exclusion of 32 studies due to reasons such as lack of reported genetic data (*n* = 17), populations not engaged in team sports (*n* = 1), and incomplete data or inadequate methodology (*n* = 14). Of the 27 studies that remained, 11 were subsequently excluded from the synthesis because they presented redundant data, non-comparable methodology, or insufficient information for result extraction. In the end, 16 studies were included in the systematic review analysis [20,21,22,23,24,25,26,27,28,29,30,31,32,33,34,35] (Figure 1).

Study selection was carried out independently by two reviewers, who worked in parallel and documented the selection process in a shared database. Any discrepancies between the two reviewers were discussed and resolved by consensus. Blinding of reviewers to authors, journal of publication, or reported results was not applied due to the nature of the field and the public accessibility of the information.

Data extraction from the included studies was performed using a standardized form designed to ensure consistency and comparability of the collected information. This form included the following categories: bibliographic information (author[s], year of publication, and country where the study was conducted), athletic level of participants (amateur vs. elite athletes), genes and polymorphisms analyzed, genotyping methods, physiological performance parameters, main findings, and the direction of the association between genotypes and performance.

The collected data were compiled and are presented in Table 2. The data were used to highlight patterns of association between genetic markers and physical performance in team sports. This stage facilitated comparative analysis across studies and the identification of methodological or conceptual variations relevant to interpreting the results.

### 2.6. Assessment of Methodological Quality

To evaluate the methodological quality of the studies included in this systematic review, the Q-Genie tool (Quality of Genetic Association Studies) was used, which was specifically developed for studies investigating associations between genetic variables and phenotypic traits [36]. This tool provides a solid framework for assessing genotype–phenotype association studies by evaluating 11 key domains related to study design, execution, and reporting (D1–D11). Specifically, D1 refers to the clarity of the research question; D2—appropriateness of the study design; D3—characteristics of the studied population; D4—participant selection method; D5—exposure assessment (genotyping); D6—outcome assessment (physical performance); D7—control of confounding factors; D8—statistical analysis; D9—interpretation of results; D10—relevance of conclusions; and D11—completeness of data reporting (Table 3).

Each evaluation criterion was scored on a scale from 1 (poor) to 7 (excellent), resulting in a total score ranging from 11 to 77. According to the Q-Genie guidelines, a score ≤ 35 indicates low study quality, a score between 36 and 75 indicates moderate quality, and a score > 45 indicates high quality [36].

All studies included in this systematic review obtained total scores above 45 points, placing them in the high-quality category according to the Q-Genie tool [36]. These results indicate that the studies were generally well-designed, with adequate descriptions of the study populations, robust genotyping methods, and relevant statistical analyses. The control of confounding variables and clear reporting of results were strengths in most of the studies. This evaluation supports the robustness of the conclusions formulated in the review and provides a high level of confidence in the validity of the genetic associations presented.

### 2.7. Registration and Protocol

This systematic review was not registered in PROSPERO or any other platform. A protocol was not prepared.

## 3. Results

Following the systematic selection process, a total of 16 studies that met the established eligibility criteria were included. These studies were analyzed in terms of the relationship between polymorphisms in specific candidate genes and relevant physical performance parameters in the context of team sports. The results were synthesized based on the characteristics of the populations investigated, the genes analyzed, the types of team sports targeted, and the identified patterns of genetic association.

### 3.1. Included Studies and General Characteristics

The 16 included studies were published between 2018 and 2025 and originated from countries such as Spain, Poland, Brazil, Italy, and others. Most of these studies focused on athletes practicing team sports such as football, handball, basketball, rugby, hockey, and volleyball, and included both professional athletes and elite-level juniors.

The studies primarily investigated the *ACTN3* and *ACE* genes, but also examined other relevant markers such as *PPARA*, *IL6*, *BDKRB2*, *FAAH*, *AGT*, *HIF1A*, and *NOS3*, which are involved in the regulation of energy metabolism, muscle contractility, inflammatory response, or injury risk. The detailed characteristics of these studies are presented in Table 2 (Section 2.5).

Sample sizes varied considerably, ranging from pilot studies to research involving hundreds of active athletes competing in national or international leagues. Most studies employed PCR-based genotyping methods and assessed objective physical performance parameters such as speed, explosive strength, aerobic endurance, intermittent effort capacity, and the rate of muscular injuries.

### 3.2. Candidate Genes and Association with Physical Performance

The most frequently investigated genes in the included studies were *ACTN3* and *ACE*, both of which were repeatedly associated with key physical performance parameters such as explosive strength, speed, and endurance. Other genes of interest included *PPARA*, *HIF1A*, *AGT*, *BDKRB2*, *FAAH*, and *IL6*, which are involved in metabolic adaptation, inflammation, recovery, and injury susceptibility. In addition to the previously discussed markers, several other candidate genes have been identified as relevant to athletic performance and predisposition to injuries. These include *VEGF*, *COL5A1*, *MCT1*, *HFE*, *COMT*, *CD36*, and *NOS3*, each contributing to a more nuanced understanding of how genetic profiles may influence physical capacity and injury risk in the context of sports.

#### 3.2.1. *ACTN3* Gene

The R577X polymorphism in the *ACTN3* gene was consistently associated with performance in power and speed-based sports. The RR genotype was more frequently found in athletes occupying positions that require explosive effort (e.g., defenders and forwards in football), while the XX genotype was correlated in some studies with enhanced endurance capacity. Studies conducted in basketball, volleyball, and handball supported these associations.

#### 3.2.2. *ACE* Gene

The I/D (insertion/deletion) polymorphism in the *ACE* gene was linked to different physiological profiles: the I allele was associated with endurance sports performance, while the D allele was related to anaerobic, sprint-type, or explosive activities. Studies in football and rugby identified these associations based on player position.

#### 3.2.3. Other Candidate Genes

Other studies analyzed genetic markers such as the following:*PPARA* and *HIF1A*—involved in mitochondrial adaptation and lipid metabolism, relevant for mixed-type sports;*BDKRB2* and *IL6*—linked to vasodilation, inflammation, and post-exercise recovery;*FAAH*—associated with the regulation of the endocannabinoid system and physiological stress;*AGT*—involved in blood pressure regulation, important for sustained effort;*NOS3* (*eNOS*)—associated with endothelial function and blood flow regulation, with potential implications for aerobic performance and injury susceptibility;*VEGF*—plays a key role in angiogenesis, relevant to both aerobic capacity and tissue integrity;*COL5A1*—linked to collagen structure and predisposition to musculoskeletal injuries;*MCT1*—involved in lactate transport, impacting tolerance to high-intensity effort;*HFE*—regulates iron homeostasis, potentially affecting recovery and endurance;*COMT*—participates in neurocognitive processes and has been associated with concussion susceptibility;*CD36*—investigated for its role in susceptibility to non-contact tissue injuries;Polygenic approaches, such as the Total Genetic Score (TGS) or panels of multiple SNPs, have been employed to integrate the cumulative contribution of multiple genetic markers to athletic performance and injury risk prediction.

Table 4 summarizes the candidate genes investigated, the phenotypes evaluated, and the direction of observed associations.

Table 4 presents a relevant synthesis of the most frequently investigated candidate genes in team sports in the context of physical performance. Each of these genes is analyzed in direct relation to its specific biological functions and the associated polymorphisms. Through this approach, the identified correlations between these genetic variations and relevant motor, metabolic, or neurophysiological parameters are outlined in relation to athletic performance activity. In this context, the presented data may help identify recurring trends in the exploration of performance genetics, contributing to a clearer conceptual framework regarding the connection between the genome and the expression of physical abilities within the specific demands of team sports.

### 3.3. Candidate Genes and Their Distribution Across Team Sports

The studies included in this systematic review cover a variety of team sports such as football, handball, basketball, volleyball, rugby, and hockey. These sports are characterized by complex physiological and tactical demands, making them relevant for investigating genetic associations with physical performance. The most frequently studied sports discipline was football, appearing in more than half of the selected studies. It was followed by other sports, all of which involve high-intensity intermittent actions, decision-making factors, and mixed energy profiles. The studies showed a tendency to correlate specific genotypes with playing positions, intermittent effort capacity, and injury rates.

Additionally, a thematic trend can be observed in the selection of genes analyzed according to the type of sport. For example, in speed- and power-based sports (handball, rugby), the focus was on the *ACTN3 R577X* polymorphism; in sports with a sustained aerobic profile (football, hockey), the association of the I allele from *ACE* with physical performance was frequently investigated; and in contact sports or those with a higher risk of muscular injuries (rugby, football), genes such as *IL6*, *AGT*, and *FAAH* were often targeted (Table 5).

This thematic distribution reflects researchers’ growing interest in the individualization of training load and injury prevention based on athletes’ genetic profiles. Correlating genotypes with the specific demands of each sport and the tactical role of players opens promising perspectives for the practical use of genetic testing within high-performance teams.

The data presented in Table 5 offer an integrated representation of the reviewed literature concerning the distribution of candidate genes according to the team sports investigated. This reflects a thematic adaptation trend in genetic research to the physiological and tactical particularities of each sports discipline. Team sports involve complex and varied metabolic demands, and the selection of genetic markers analyzed appears to have been guided by these specific characteristics. This differentiated approach supports the potential of applied genetics in elite sports for optimizing selection, personalizing training, and preventing injuries. The results highlight the relevance of an integrated analysis in which the genetic profile is correlated with the specific physical demands and role-related requirements of each sport.

### 3.4. Genetic Predisposition to Injuries

A recurrent topic in the reviewed scientific literature is the association between genetic profile and the predisposition to muscle or musculoskeletal injuries in team sports. This research direction primarily targets sports characterized by high-intensity intermittent effort, in which athletes are frequently exposed to trauma, physical contact, and repetitive biomechanical stress (Table 6).

Several studies have investigated the link between polymorphisms in the *IL6*, *AGT*, *BDKRB2*, and *FAAH* genes and injury risk or recovery time. For example, *IL6* (rs1800795) is involved in regulating the inflammatory response and may influence the speed of tissue regeneration after exertion, while *AGT* (rs699) and *BDKRB2* (rs5810761) are associated with blood pressure control, vascular function, and susceptibility to micro muscle injuries. The rs324420 polymorphism in *FAAH* has been analyzed in the context of physiological stress regulation and pain perception, with potential implications for tolerance to post-training discomfort and injury.

At the same time, other genetic markers relevant to connective tissue integrity and endothelial function have been reported. For instance, *COL5A1* (rs12722) has been associated with collagen resistance and an increased risk of musculoskeletal injuries, while *NOS3* (rs1799983, Glu298Asp) influences endothelial function and blood flow, being correlated with injury susceptibility and aerobic performance. In addition, the *VEGF* (rs2010963, −634C > G) variant has been linked to angiogenesis and injury vulnerability, whereas the *COMT* (rs4680, Val158Met) polymorphism has been reported in relation to an increased risk of concussions in rugby athletes.

Additionally, two of the included studies [27,28] employed a polygenic approach, calculating a composite genetic score (TGS—Total Genotype Score) to estimate the risk of muscle injuries in elite football players. These approaches suggest a promising direction for the application of genetics in sports practice, with the aim of preventing injuries through personalized interventions.

The available results indicate a moderately consistent association between genetic profile and injury vulnerability, but also highlight the need for larger sample sizes and the integration of other risk factors (training load, playing position, age, and medical history).

### 3.5. General Synthesis of Genetic Associations

The analysis of studies included in this systematic review offers a comprehensive view of how genetic variations may influence physical performance in team sports, while also suggesting that there is no linear or singular relationship between a specific gene and a particular type of performance. Instead, a multifactorial and polygenic model emerges, in which polymorphisms act cumulatively, in interaction with environmental factors, training strategies, and the competitive context. The convergence of multiple studies on recurring genes indicates that certain biological pathways are central to explaining variability in physiological responses—namely, energy metabolism, muscle contractility, cardiovascular adaptation, tissue recovery, and inflammatory response. These factors are essential in team sports disciplines, where performance involves rapid alternations between high-intensity efforts and partial recovery periods, coordination, and decision-making capacity under stress. An emerging methodological trend can also be observed: a shift from single-gene studies to multigenic models that consider multiple relevant polymorphisms. This holistic approach, expressed through cumulative genotype scores, increases predictive accuracy and enables the integration of genetic information into functional models that better reflect real-world sports conditions.

The distribution of sports disciplines investigated and the variety of performance parameters analyzed highlight considerable contextual diversity across studies, which hinders direct comparison of results but still provides a valuable resource for identifying common patterns. This heterogeneity reflects the nature of team sports, where player roles, intensity of demands, and training types can vary significantly within the same sports discipline.

Overall, current evidence supports the idea that an athlete’s genetic profile can provide relevant, but not exclusive, information for understanding performance. Leveraging this profile requires not only the identification of markers with associative value, but also their contextualization based on playing roles, training objectives, and the athlete’s individual history. Therefore, the practical applicability of these genetic data requires an integrated, interdisciplinary, and ethically grounded approach. Clearly, the available studies provide a strong foundation for developing decision-support tools in sports contexts. However, challenges remain in clinically validating these associations, standardizing methodologies, and building large-scale databases segmented by sport type and performance level.

In conclusion, the general synthesis of current findings supports a transition from descriptive to applied and functional genetics, in which the analysis of polymorphisms becomes part of a broader ecosystem for sports optimization. These summarized findings provide a necessary framework for critically interpreting results in light of the current research landscape in sports genetics.

## 4. Discussion

This systematic review aimed to synthesize the existing evidence regarding associations between specific genetic polymorphisms and physical performance in team sports, focusing both on the convergence of findings across studies and the practical relevance of these discoveries in sports practice. The analysis was based on 16 studies that examined the role of genetic variations in the context of team sports, an area less explored compared to individual sports.

The findings of our study indicate that, in team sports, there is a growing interest in identifying genetic associations that can support physical performance and injury prevention. Moreover, the results confirm that the *ACTN3 R577X* and *ACE I/D* polymorphisms remain among the most relevant genetic predictors of physical performance, even in the face of the complex and multidimensional demands characteristic of team-based disciplines. These findings provide a solid foundation for understanding how genetic variations may influence physiological responses to training, intermittent effort capacity, injury risk, and even psychological adaptation to the specific demands of team sports.

### 4.1. Interpretation of Results in the Context of Team Sports

The vast majority of the studies analyzed in this systematic review reflect a convergent scientific interest in key candidate genes with proven physiological relevance in team sports. Among these, *ACTN3* and *ACE* stand out as the most frequently investigated, due to their contribution to interindividual variability in strength, speed, endurance, and recovery capacity—essential attributes in disciplines characterized by intermittent physical demands, specific to team sports [20,21,23,25,26,29,30]. In particular, the *ACTN3 R577X* polymorphism has been consistently associated with differences in physical capacities among athletes: the RR genotype (functional) has been correlated with superior performance in power and speed-based activities, while the XX genotype (non-functional) is associated with greater metabolic efficiency and better tolerance to sustained effort, especially in positions that require prolonged aerobic activity [5]. According to these studies, athletes with RR or RX genotypes achieved superior results in tests such as sprinting, vertical jumps, and explosive strength exercises, a trend confirmed by data from sports like handball, basketball, rugby, and football [20,30]. This functional relationship is further supported by other studies [37,38,39]. For example, Garatachea et al. (2014) highlighted a clear association between *ACTN3 R577X* and lower-limb explosive power in basketball players [37]. Similarly, Heffernan et al. (2016) demonstrated a differentiated distribution of this polymorphism depending on playing position in rugby athletes, suggesting a functional integration of the genetic profile into tactical demands [38]. In the same vein, Yang et al. (2023) confirmed the relevance of *ACTN3* in distinguishing performance levels among young football players in China, emphasizing its potential for early talent identification [39]. Moreover, recent research by Konopka et al. (2023) supports these observations in mixed-performance contexts, demonstrating the validity of *ACTN3* as a biomarker in sports such as basketball and football [40], while the meta-analysis by Ma et al. (2013) reported a higher prevalence of RR/RX genotypes in elite athletes from power sports, reinforcing the robustness of these associations [41]. On the other hand, despite the dominant perspective emphasizing the advantage of functional genotypes, some studies [5,6,7,42] have suggested that the XX genotype may provide metabolic benefits and superior adaptability to long-duration effort, offering strategic utility in positions such as midfielders or central defenders, where sustained intermittent effort and quick decision-making are essential. This approach opens new directions for customizing player roles and training based on the athlete’s genetic profile.

Regarding the *ACE I/D* polymorphism, the results of our review support the idea that the I allele is correlated with intermittent endurance sports, while the D allele is associated with high-intensity anaerobic activities [20,26,29]. This division is also supported by other studies [43,44,45]. For instance, McAuley et al. (2020), in a meta-analysis focused on football, confirmed this functional relationship [43], while de Almeida et al. (2022) highlighted the increased predictive value of the *ACTN3*–*ACE* combination for the prevention of non-contact muscle injuries [44]. Furthermore, Gonzalez-Garcia & Varillas Delgado (2024) demonstrated that genetic variation influences performance throughout a competitive season, in correlation with accumulated fatigue and individual neuromuscular capacity [45].

At the mechanistic level, the D allele has been associated with higher levels of angiotensin-converting enzyme (ACE) activity, which may lead to increased muscle mass and elevated blood pressure, offering an advantage in contact sports such as rugby and hockey [21,26]. On the other hand, the I allele is linked to lower enzymatic activity but greater cardiovascular efficiency and a better metabolic response to prolonged efforts, making it more favorable in sports or positions with an aerobic profile [3,46].

This dual relationship was supported by a comprehensive meta-analysis by Ma et al. (2013), which found increased frequencies of the D allele among strength athletes and the I allele in endurance sports [41]. Additionally, Myerson et al. (1999) showed that elite endurance athletes exhibited a higher prevalence of II and ID genotypes, suggesting more efficient cardiovascular adaptation [47].

The relevance of the *ACE I/D* polymorphism to team sports is emphasized by the mixed nature of these disciplines, where physiological demands involve both anaerobic components (such as explosive power, sprinting, or jumping) and aerobic components (intermittent effort, rapid recovery capacity). In this context, the distribution of ACE alleles may influence the optimal physiological response depending on the role played on the field: the D allele favors muscle hypertrophy and short-term strength, while the I allele is associated with increased cardiovascular efficiency and tolerance to prolonged effort. This genetic versatility offers strategic potential in personalizing training and athlete selection, based on playing position and the tactical demands of the game [48].

Expanding the analysis beyond the *ACTN3* and *ACE* genes, several studies included in this review have investigated additional genetic markers with complementary potential that are involved in key physiological processes essential for performance in team sports, such as energy metabolism, adaptation to hypoxia, inflammatory response, endothelial function, and pain perception. Genes such as *PPARA*, *HIF1A*, *IL6*, *NOS3*, *AGT*, *BDKRB2*, and *FAAH* have been associated with traits like aerobic capacity, metabolic efficiency, fatigue resistance, and injury susceptibility [25,26,29,30]. A notable example is the *PPARA rs4253778* polymorphism, involved in fatty acid oxidation, which has been linked to a favorable endurance profile, particularly among athletes in mixed-type sports such as basketball or hockey [49,50,51]. Other studies confirmed these associations, highlighting the prevalence of the GG genotype in endurance athletes and the association of CC genotypes with greater strength and muscle mass, suggesting a potential bidirectional effect of this gene [52,53].

Regarding cardiovascular response and adaptation to high-intensity effort, the *HIF1A* and *NOS3* genes have been associated with oxygen transport efficiency and endothelial function, influencing performance in intermittent effort contexts [54,55,56]. For instance, Eynon et al. (2010) reported that the *HIF1A Pro582Ser* polymorphism is linked to enhanced performance in endurance sports, while the *G894T* polymorphism in *NOS3* has been associated with a more effective vasodilatory response and improved recovery capacity [50].

Moreover, recent approaches emphasize multigenic analysis, integrating the cumulative effects of multiple genetic variations. Hall et al. (2021) demonstrated that a set of polymorphisms associated with VO_2_max adaptation contributes to performance differentiation in both rugby and marathon running [57]. In a complementary direction, Balberova et al. (2021) proposed a functional gene network involved in contractility, metabolism, and muscle adaptation, providing a theoretical framework for linking genetic profiles with the physiological demands of team sports [58]. Additionally, Gonzalez-Garcia & Varillas-Delgado (2024) highlighted that the genetic profile influences the progression of neuromuscular performance throughout the season in football players, emphasizing the dynamic interaction between genetic predisposition and training stimuli [45].

In our study, several works explored cumulative genotype scores (Total Genotype Score—TGS) as an integrative method for assessing injury risk and adaptation capacity to physical effort in team sports. For instance, Maestro et al. (2022) and Massidda et al. (2024) proposed models based on the integration of multiple genetic variants into predictive scores for muscle injuries, suggesting a clear shift from monogenic approaches toward more complex polygenic models, better reflecting the multifactorial nature of athletic performance [27,28].

This trend is supported by synthesis studies such as Ahmetov & Fedotovskaya (2015), who state that sports performance is determined by the interaction of dozens or even hundreds of genes, each with a modest but cumulative effect [7]. In the same direction, Pranckeviciene et al. (2021) showed that TGS (Total Genotype Score) derived from *ACTN3*, *ACE*, and *IGF1* polymorphisms can distinguish elite athletes from amateurs based on physical fitness level and injury susceptibility [59]. Similarly, Sarzynski et al. (2016) demonstrated that multi-gene scores can more effectively predict maximal effort performance and individual response to training, especially in sports with mixed physiological profiles [60]. Moreover, Varillas-Delgado et al. (2022) highlighted the utility of cumulative scores in the context of athlete selection and preparation in team sports, where demands are variable and adaptive [4]. This integrative approach suggests that individual genotypes should not be analyzed in isolation, but rather as part of a multigenic predictive network, which can be leveraged for personalized training and injury prevention.

In addition to those already discussed, associations have also been reported for other candidate genes, such as *VEGF* and *COL5A1*, which are involved in angiogenesis and collagen integrity. These findings support the hypotheses that vascular adaptation and connective tissue structure may contribute both to aerobic performance and to the risk of musculoskeletal injuries [27,31]. Similar observations have previously been described in the synthesis literature on the sports genome [7]. Other variations, such as those in *MCT1* and *HFE*, have been associated with tolerance to high-intensity exercise and recovery capacity, complementing existing evidence on the role of lactate metabolism and iron homeostasis in sports performance [31]. Semenova et al. (2023) also highlighted the relevance of these metabolic pathways in sports with intermittent demands [1].

In contact sports, the *COMT rs4680* polymorphism has been correlated with susceptibility to head trauma and elite athlete status [34], while *CD36 rs1761667* has been associated with the risk of non-contact injuries [32]. These findings expand the classical picture dominated by *ACTN3* and *ACE* and align with the perspective of Eynon et al. (2010), who emphasized the importance of neurocognitive and inflammatory markers in intermittent sports [50]. At the same time, the use of polygenic approaches, such as the Total Genotype Score (TGS) and Multi-SNP panels, indicates a shift from assessing individual markers to integrated genetic models, which offer greater applicability for predicting both performance and injury risk [28,35]. This direction is further supported by other studies proposing a transition towards a polygenic framework in sports genetics [4,7].

This systematic review confirms that team sports provide an optimal framework for investigating the contribution of genetic profiles to athletic performance, due to their physiological, cognitive, and tactical complexity. Current evidence supports reproducible associations between polymorphisms such as *ACTN3 R577X* and *ACE I/D* and relevant performance traits (strength, speed, endurance, and recovery), with functional differences depending on playing position and the specific characteristics of each sport. Furthermore, the integrative analysis of the included studies suggests that polygenic and personalized approaches offer real potential for improving strategies related to athlete selection, injury prevention, and training.

Despite some methodological and population-based variability, the consistency of associations across studies reinforces the validity of using genetic profiles as a complementary tool in sports science. However, the application of this information in practice must be accompanied by rigorous scientific validation and adherence to ethical standards, in order to prevent abusive or deterministic use of genetic data. Overall, the findings of this review support the responsible integration of genetics into performance optimization in team sports, as part of a multidimensional and evidence-based approach.

### 4.2. Consistencies and Discrepancies with the Scientific Literature

The results of this systematic review align with the prevailing directions in the international literature on sports genetics, confirming the significant role of certain polymorphisms in predisposing individuals to athletic performance. In particular, there is thematic continuity between the included studies and previous theoretical syntheses, which identify *ACTN3* and *ACE* as core markers in the assessment of athletic profiles [2,5,7,51]. Recent works, such as those by Varillas-Delgado et al. (2022) and Roth (2021), expand on this perspective, highlighting the applicability of genetic information not only for initial athlete selection but also for decisions related to field positioning and training planning [4,12].

Moreover, there is a clear convergence between the international literature and the present review regarding the trend toward multigenic and integrative approaches. Studies by Sarzynski et al. (2016) and Heffernan et al. (2016) support the use of cumulative scores (TGS) for a more realistic prediction of performance and injury risk [38,60]. These directions reflect a paradigm shift from the isolated analysis of a single gene to the development of more robust predictive models.

On the other hand, some discrepancies reported in the literature relate to the variability in study design and characteristics of the populations investigated. For example, in the meta-analysis conducted by McAuley et al. (2020) on *ACTN3* and *ACE* polymorphisms in football, differences were noted based on sex, age, competitive level, and ethnicity—all of which significantly influenced the results [43]. Such contextual factors may contribute to inconsistencies in the statistical significance of genetic associations.

Another critical issue is the lack of standardization in defining athletic performance. The reviewed studies include highly variable parameters—from objective laboratory indicators (e.g., VO_2_max, isometric strength) to competitive scores or self-assessments—which makes direct comparison between studies difficult. Moreover, many investigations focus exclusively on male athletes, excluding potential sex- or age-dependent effects [56,59].

In conclusion, although there is general agreement regarding the relevance of genetics in team sports, the current literature also highlights methodological limitations that affect the reproducibility of results. These findings emphasize the need for longitudinal, multicenter studies that integrate genetics, training context, and physiological response within a functional, scientifically validated predictive framework.

### 4.3. Implications for Sports Practice

The results of this study, complemented by the comparison with international specialized literature, highlight an important practical potential for integrating the genetic profile in the optimization of sports performance, injury prevention, and the personalization of training interventions in team sports. Specifically, genetic information can contribute to positional selection of athletes, according to their predisposition to strength, speed, or endurance; to the individualization of training, in accordance with the profile of metabolic, neuromuscular, or cardiovascular adaptation; to the planning of recovery periods and the anticipation of muscular risks, especially in the case of athletes prone to injuries; to the optimization of the selection process in training centers, allowing the completion of physical and psychological assessments with an objective and non-invasive profile; and to the monitoring of young athletes’ development, in correlation with biological maturation and individual responses to training stimuli. However, the practical implementation of these findings must be approached with caution and ethical discernment, considering the lack of full consensus on the interpretation of genetic scores at the individual level and the high variability between populations and sports disciplines, as well as the risk of stigmatization or incorrect selection of athletes in the absence of an integrative assessment (genetic + physiological + psychological).

Therefore, the role of the genetic profile must be understood as a complementary one, not a decisive one, in the equation of sports performance. It only becomes relevant when correlated with the training context, style of play, competition load, and the psychosocial environment of the athlete.

### 4.4. Limitations

Although this systematic review provides a comprehensive overview of relevant genetic associations in team sports, there are several methodological and contextual limitations that must be acknowledged in order to correctly understand the scope and applicability of the conclusions.

A first limitation of this study is given by the heterogeneity of the included research. The investigated populations vary significantly in terms of the type of sport practiced (football, handball, basketball, rugby, hockey, etc.), level of performance (amateur vs. professional athletes, juniors vs. seniors), and geographical or ethnic background. These differences may influence both the expression of the analyzed genes and the practical relevance of the investigated polymorphisms. Another limitation is that most of the studies are cross-sectional and do not allow for the establishment of causal relationships. The lack of longitudinal studies that track genetic adaptations over time or in response to training limits the predictive power of the results. Also, a few studies applied randomized designs or used well-defined control groups. Moreover, the variation in the way physical performance is measured represents another important limitation of this study. The parameters used (speed, strength, endurance, and injuries) are not uniform, and the lack of standardized evaluation tools creates significant difficulties in making direct comparisons between studies. Additionally, a few of the studies analyzed addressed the multifactorial dimension of performance, namely the integration of genetic factors with epigenetic, nutritional, psychological, or environmental ones. This fragmented approach may lead to overestimating the influence of a single gene while ignoring the complex interactions among multiple biological levels. Language and accessibility barriers also represent a limitation of this study, as only studies published in English and indexed in certain scientific databases were included in the analysis. In this regard, there is a possibility that some relevant research may have been omitted.

These limitations do not invalidate the results of this study. They merely suggest that these results should be interpreted with methodological caution and placed within the context of a research field that is still developing.

## 5. Conclusions

This systematic review synthesized the evidence presented in the 16 analyzed studies regarding the relationship between certain genetic polymorphisms and physical performance in team sports, highlighting the relevance of markers such as *ACTN3 R577X* and *ACE I/D*. These two polymorphisms have proven to be the most consistent genetic predictors of essential motor qualities—strength, speed, endurance, and recovery capacity in complex sporting contexts characterized by intermittent demands and diverse tactical roles. At the same time, the inclusion of complementary genes such as *PPARA*, *IL6*, *NOS3*, *AGT*, *FAAH*, and *BDKRB2* in the analysis allowed for a broader understanding of the applicative potential of genetic profiling, particularly in relation to adaptation to effort, injury susceptibility, and inflammatory/metabolic response.

Another area of interest is the shift toward polygenic models and cumulative scores (TGS), which offer a more realistic and integrative approach to athletic performance. These reflect a necessary transition from monogenic interpretations to a systemic understanding that can support the planning of personalized interventions in sports training, athlete selection, and injury prevention.

Nevertheless, methodological discrepancies, variations in the populations analyzed, non-uniform definitions of performance, and the lack of longitudinal validation limit the direct applicability of these data in sports practice. It is essential that future research be conducted within rigorous multicenter protocols, with diversified samples, standardized assessments, and careful ethical integration of genetic information.

Therefore, the integration of genetics into sports science should not be seen as a substitute for traditional evaluations, but rather as a complementary tool, capable of refining the understanding of individual potential and contributing to the optimal personalization of athletic development in high-level team sports.

## Figures and Tables

**Figure 1 genes-16-01079-f001:**
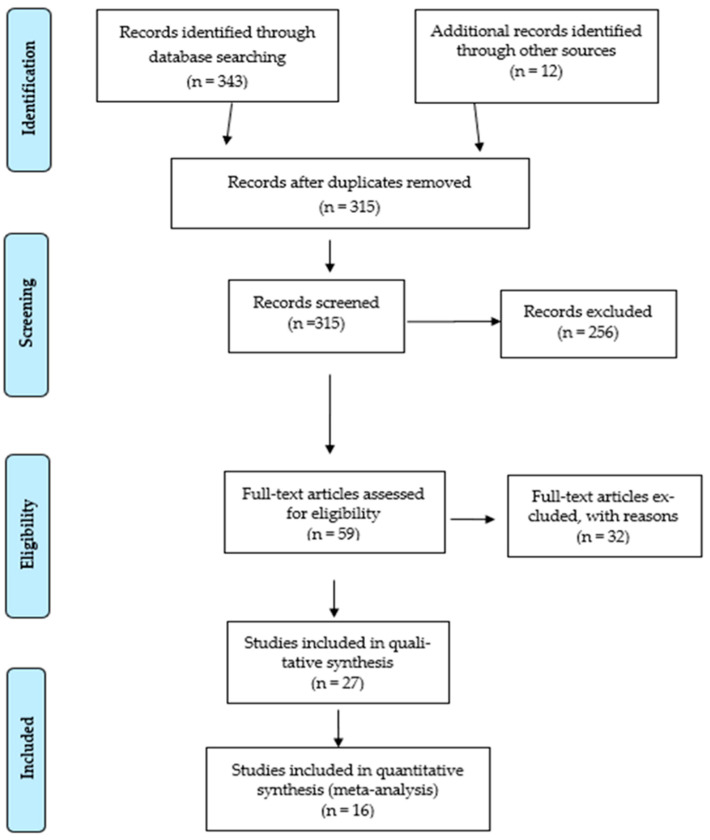
PRISMA flow diagram of study selection.

**Table 1 genes-16-01079-t001:** PICO model for study design.

P (Population):	I (Intervention):	C (Comparison):	O (Outcomes):
Athletes from team sports, both elite and amateur, are active in disciplines such as football, handball, rugby, hockey, basketball, etc.	Presence of genetic polymorphisms in candidate genes involved in physiological mechanisms essential to performance (e.g., *ACTN3 R577X*, *ACE I/D*, *PPARGC1A*, *IL6*, *NOS3*, and Brain-Derived Neurotrophic Factor (*BDNF*)).	Alternative genotypes (e.g., *ACTN3* XX vs. RR), absence of polymorphisms, athletes without a specific genetic profile, and athletes from individual or different types of sports (endurance vs. strength).	Physical performance parameters: speed, explosive strength, aerobic endurance, intermittent effort capacity, recovery time, physiological adaptation to training, and predisposition to muscular and osteoarticular injuries.

**Table 2 genes-16-01079-t002:** Main characteristics of the studies included in this systematic review.

Author(s) And Year	Country	Sport/Population	Genes Analyzed	Evaluated Parameters	Key Findings
Orysiak et al., 2018 [20]	Poland	Polish athletes (various sports)	*ACE*, *ACTN3*	Muscle composition, strength	Combined influence of *ACE* and *ACTN3* polymorphisms on muscle profile
Pasqualetti et al., 2022 [21]	Italy	Elite-level rugby players	*ACE*, *ACTN3*, Monocarboxylate Transporter 1 (*MCT1*)	Sports performance	Preliminary associations between polymorphisms and performance level
McAuley et al., 2023 [22]	-	Football players	Systematic review—multiple genes	Injury predisposition	Genetic evidence of increased injury risk in football
Lima et al., 2023 [23]	Brazil	Basketball players, Brazilian League I	*ACTN3 R577X*	Field position	Association between *ACTN3* genotype and playing position (center vs. wing)
Silva et al., 2023 [24]	Portugal	Elite athletes (sport unspecified)	*FAAH rs324420*	General performance, metabolic potential	Possible role of *FAAH* in optimizing elite performance
Altynova et al., 2024 [25]	Kazakhstan	Athletes from multiple disciplines	Multiple genetic markers (genetic passport)	General performance	An individual genetic profile may guide personalized training
Doğan et al., 2024 [26]	Turkey	Ice hockey players	ACE, ACTN3, PPARA, and Hypoxia-Inducible Factor 1-Alpha (*HIF1A*)	General physical performance	Different genotypes correlated with distinct physical outcomes
Maestro et al., 2022 [27]	Spain	High-level football players	*ACTN3*, *ACE*, Collagen Type V Alpha 1 Chain (*COL5A1*), Insulin-Like Growth Factor 2 (*IGF2*), *IL6*, Tumor Necrosis Factor (*TNF*), NOS3, and Vascular Endothelial Growth Factor A (*VEGFA*)	Types of muscle injuries, injury history	*ACTN3 XX*, *COL5A1 TT*, and *IL6 GG* genotypes are associated with increased muscle injury risk
Massidda et al., 2024 [28]	Italy	High-level football players	Total Genotype Score (*TGS*)	Muscle injuries	*TGS* correlated with the frequency of muscle injuries in football players
Vavak et al., 2025 [29]	Slovakia	Volleyball and basketball players	*ACE*, *ACTN3*, *PPARA*, *HIF1A*, and Adenosine Monophosphate Deaminase 1 (*AMPD1*)	Vertical jump	Genetic correlations with jump parameter variations
Silvino et al., 2025 [30]	Brazil	Junior handball players	*ACTN3*, Bradykinin Receptor B2 (*BDKRB2*), and *AGT*	Physical performance	Significant associations between *ACTN3* and *BDKRB2* with physical performance
La Montagna et al., 2019 [31]	Italy	Professional football players	*ACTN3*, *COL5A1*, *MCT1*, *VEGF*, and Homeostatic Iron Regulator (*HFE*)	Performance and injuries	*ACTN3* and *VEGF* are associated with performance; *COL5A1* and *HFE* are associated with injury predisposition
El Ouali et al., 2025 [32]	Morocco	Elite cyclists and hockey players	Cluster of Differentiation 36 (*CD36*) rs1761667	Susceptibility to non-contact injuries	*CD36* is associated with an increased risk of tissue injuries
Silvino et al., 2025 [33]	Brazil	Handball players	*ACE (I/D)*	Physical performance	Association between *ACE I/D* polymorphism and physical performance
Antrobus et al., 2023 [34]	UK	Elite rugby players	Catechol-O-Methyltransferase (*COMT*) rs4680	Elite athlete status, concussion risk	*COMT rs4680* is associated with elite status and predisposition to concussions
Rodas et al., 2019 [35]	Spain	Football, futsal, handball, basketball, hockey players	Multiple SNPs	Tendinopathy risk prediction	Genomic profile predictive of tendinopathy in team sports

**Table 3 genes-16-01079-t003:** Methodological quality assessment of included studies.

No	Study	D1	D2	D3	D4	D5	D6	D7	D8	D9	D10	D11	Total Score	Quality Rating
1	Orysiak et al., 2018 [20]	7	4	6	6	7	4	3	4	5	5	3	54	High
2	Lima et al., 2023 [23]	7	3	7	3	6	5	6	7	7	3	3	57	High
3	Vavak et al., 2025 [29]	3	6	3	5	3	4	7	4	6	6	3	50	High
4	Altynova et al., 2024 [25]	6	3	4	3	3	7	7	7	5	4	6	55	High
5	Pasqualetti et al., 2022 [21]	3	7	3	7	6	4	6	6	6	4	6	58	High
6	Doğan et al., 2024 [26]	6	7	6	4	6	4	4	4	5	6	5	57	High
7	Silvino et al., 2025 [30]	5	6	6	5	3	7	6	4	4	7	6	59	High
8	Silva et al., 2023 [24]	7	7	7	6	6	3	7	6	3	5	5	62	High
9	Maestro et al., 2022 [27]	6	5	6	7	7	3	7	6	4	5	7	63	High
10	McAuley et al., 2023 [22]	7	4	4	5	3	7	5	7	3	5	6	56	High
11	Massidda et al., 2024 [28]	5	7	4	5	4	4	7	4	3	7	4	54	High
12	La Montagna et al., 2019 [31]	7	6	6	7	6	5	6	7	6	6	6	68	High
13	Rodas et al., 2019 [35]	6	6	6	6	5	6	5	6	6	5	5	62	High
14	Antrobus et al., 2023 [34]	6	5	5	5	6	5	6	6	5	5	5	59	High
15	El Ouali et al., 2025 [32]	5	6	5	5	4	5	5	6	5	6	4	56	High
16	Silvino et al., 2025 [33]	6	6	6	6	6	6	6	6	5	6	6	65	High

**Table 4 genes-16-01079-t004:** Candidate genes associated with performance in team sports.

Gene	Polymorphism	Associated Physiological Parameter	Team Sports Investigated	Type of Association	Included Studies
*ACTN3*	R577X (rs1815739)	Strength, speed, and sprint	Football, handball, basketball, and volleyball	Positive association (RR with power sports; XX with endurance sports)	Orysiak et al., 2018 [20]; Lima et al., 2023 [23]; Silvino et al., 2025 [30]; La Montagna et al., 2019 [31]
*ACE*	I/D (rs1799752)	Endurance, anaerobic effort	Football, rugby, hockey	I allele → endurance; D allele → strength	Orysiak et al., 2018 [20]; Pasqualetti et al., 2022 [21]; Doğan et al., 2024 [26]; Silvino et al., 2025 [33]
*PPARA*	rs4253778	Lipid metabolism, adaptation to mixed effort	Basketball, volleyball	Functional association with mixed-type effort	Vavak et al., 2025 [29]; Doğan et al., 2024 [26]
*HIF1A*	rs11549465	Metabolic efficiency under hypoxia	Volleyball, hockey	Positive association in intermittent sports	Vavak et al., 2025 [29]; Doğan et al., 2024 [26]
*AGT*	M235T (rs699)	Blood pressure regulation, physiological stress	Handball	Association with cardiovascular adaptation	Silvino et al., 2025 [30]
*BDKRB2*	−9/+9 (rs5810761)	Vasodilation, recovery	Handball	Association with post-effort recovery time	Silvino et al., 2025 [30]
*IL6*	rs1800795 (G174C)	Inflammation, recovery	Football	Association with inflammatory response and injuries	Maestro et al., 2022 [27]; McAuley et al., 2023 [22]
*FAAH*	rs324420	Stress tolerance, neuromodulatory regulation	Elite sports (unspecified)	Theoretical association with general physiological response	Silva et al., 2023 [33]
*NOS3* (*eNOS*)	Glu298Asp (rs1799983)	Endothelial function, blood flow, and muscle injuries	Football	Association with aerobic performance and injury susceptibility	Maestro et al., 2022 [27]
*COL5A1*	rs12722	Collagen integrity, injury risk	Football	Association with muscle injuries	La Montagna et al., 2019 [31]
*MCT1*	rs1049434	Lactate transport, exercise tolerance	Football	Association with tolerance to high-intensity exercise	La Montagna et al., 2019 [31]
*HFE*	C282Y and others	Iron homeostasis	Football	Possible impact on recovery and performance	La Montagna et al., 2019 [31]
*COMT*	Val158Met (rs4680)	Neurotransmission, risk of head trauma (concussions)	Rugby	Association with elite status and concussions	Antrobus et al., 2023 [34]
*CD36*	rs1761667	Lipid metabolism, injury risk	Field hockey	Association with non-contact injuries	El Ouali et al., 2025 [32]
*VEGF*	−634 C > G (rs2010963)	Angiogenesis, performance, and injuries	Football	Association with performance and injuries	La Montagna et al., 2019 [31]
*TGS* (polygenic)	—	Composite genetic score	Football	Correlated with injuries	Massidda et al., 2024 [28]
Multi-SNPs (panel)	—	Tendinopathy prediction	Football, futsal, basketball, handball, and hockey	Predictive genomic profile	Rodas et al., 2019 [35]

**Table 5 genes-16-01079-t005:** Distribution of candidate genes by team sport discipline/investigated studies.

Team Sport Discipline	No. of Studies	Analyzed Population	Genes Studied
Football	6	Junior and professional athletes	*ACTN3*, *ACE*, *IL6*, *AGT*, *NOS3*, *COL5A1*, *MCT1*, and Multi-SNP panel
Handball	3	Junior and performance athletes	*ACTN3*, *BDKRB2*, *AGT*, *ACE*, and Multi-SNP panel
Basketball	3	First league players (Brazil), performance athletes	*ACTN3*, *PPARA*, and Multi-SNP panel
Volleyball	2	Performance athletes, mixed volleyball	*ACTN3*, *PPARA*, and *HIF1A*
Rugby	2	Elite athletes (Italy, UK)	*ACE*, *ACTN3*, *MCT1*, and *COMT*
Hockey (ice and field)	3	National athletes (Turkey, Morocco), elite athletes	*ACE*, *HIF1A*, *PPARA*, *CD36*, and Multi-SNP panel
Futsal	1	Elite athletes	Multi-SNP panel

**Table 6 genes-16-01079-t006:** Genes and polymorphisms associated with injury risk.

Gene	Polymorphism	Associated Risk	Source Study(ies)
*IL6*	rs1800795	Increased inflammation, delayed recovery	Maestro et al., 2022 [27]; McAuley et al., 2023 [22]
*AGT*	rs699 (M235T)	Vascular susceptibility, mechanical stress	Silvino et al., 2025 [30]
*BDKRB2*	rs5810761 (−9/+9)	Risk of micro muscle injuries	Silvino et al., 2025 [30]
*FAAH*	rs324420	Low pain tolerance, physiological stress	Silva et al., 2023 [24]
*NOS3* (*eNOS*)	rs1799983 (Glu298Asp)	Predisposition to muscle injuries via endothelial dysfunction	Maestro et al., 2022 [27]
*COL5A1*	rs12722	Collagen integrity, increased risk of musculoskeletal injuries	La Montagna et al., 2019 [31]
*VEGF*	rs2010963 (−634C > G)	Injury susceptibility, influence on angiogenesis	Maestro et al., 2022 [27]; La Montagna et al., 2019 [31]
*HFE*	C282Y and other variants	Possible impact on recovery and injury susceptibility	La Montagna et al., 2019 [31]
*COMT*	rs4680 (Val158Met)	Increased risk of concussions (head trauma)	Antrobus et al., 2023 [34]
Multi-SNP panels	A combination of SNPs from multiple candidate genes	Predictive genomic profile for tendinopathy and injuries	Rodas et al., 2019 [35]
Composite Genotype (TGS)	A combination of polymorphisms from multiple genes	Cumulative score associated with the risk of muscle injuries	Massidda et al., 2024 [28]; Maestro et al., 2022 [27]

## Data Availability

No new data were created or analyzed in this study. Data sharing is not applicable to this article.

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
