# Peer review of "Physical Performance and Sports Genetics: A Systematic Review of Candidate Gene Polymorphisms Involved in Team Sports"

_genes, 2025, doi:10.3390/genes16091079_

Round 1

Reviewer 1 Report

Comments and Suggestions for Authors

During the systematic review, the authors analyzed data from eleven different papers published between 2018 and 2025, focusing on the potential role of genomic analysis in physical performance and sports genetics. This is an important review in the field of gene polymorphisms in sports. However, several relevant studies that could strengthen the findings on certain polymorphisms were not included, such as La Montagna et al., JCP 2020. I suggest expanding the number of papers analyzed in the review to enhance its significance and ensure that important information is not overlooked.

This would help increase the reliability of the review in the context of DNA polymorphism and sports.

The inclusion of other papers is not a simple reference; it represents a broader and deeper analysis of the available data.

Overall, the review is clear in terms of the introduction, results, and discussion, and I don’t see a need to modify those sections. The figures and tables are also appropriate and well presented.

Author Response

Dear Reviewer,

We would like to sincerely thank you for your detailed review and for the extremely valuable comments you provided. We greatly appreciated both the constructive tone of your feedback and your specific suggestion to expand the corpus of included papers, which proved essential for increasing the consistency and relevance of our work. All modifications have been highlighted in blue to facilitate your consultation.

Your comment:

“During the systematic review, the authors analyzed data from eleven different papers published between 2018 and 2025, focusing on the potential role of genomic analysis in physical performance and sports genetics. This is an important review in the field of gene polymorphisms in sports. However, several relevant studies that could strengthen the findings on certain polymorphisms were not included, such as La Montagna et al., JCP 2019. I suggest expanding the number of papers analyzed in the review to enhance its significance and ensure that important information is not overlooked. This would help increase the reliability of the review in the context of DNA polymorphism and sports. The inclusion of other papers is not a simple reference; it represents a broader and deeper analysis of the available data. Overall, the review is clear in terms of the introduction, results, and discussion, and I don’t see a need to modify those sections. The figures and tables are also appropriate and well presented.”

Response:

We carefully followed your very pertinent suggestion and expanded the set of studies included in our systematic review. Thus, the number of papers has increased from 11 to 16, by integrating additional relevant articles published between 2018–2025, including La Montagna et al., J Cell Physiol, 2019, as you kindly suggested.

  • These additions are reflected in Figure 1 (PRISMA flow diagram) and in Table 1 (main characteristics of included studies).
  • Their methodological quality has been evaluated and reported in Table 2 (Q-Genie).
  • Additional genetic markers (such as NOS3, VEGF, COL5A1, MCT1, HFE, COMT, and CD36) have been integrated into Section 3.2 Candidate genes and association with physical performance (lines 334-338, 359-371) and into Table 3 and Table 4.
  • Data regarding injury predisposition have been further strengthened in Table 5 and in lines 431-438.
  • Finally, we expanded the Discussion section (lines 615-635), providing comparative analysis between the newly included articles and the pre-existing literature, which we believe led to a more rigorous and nuanced interpretation of the evidence.

We are confident that your suggestion has significantly improved our manuscript, giving it greater depth and robustness. Once again, we would like to warmly thank you for your valuable recommendation, which has elevated the overall quality of our review and strengthened the validity of our conclusions.

Reviewer 2 Report

Comments and Suggestions for Authors

The manuscript "Physical performance and sports genetics: A systematic review of candidate gene polymorphisms involved in team sports" in a systematic way summarizes findings of 11 articles on genetic markers that play important role in teams sports. While this information is very useful in practice, it lacks the rigor of the systematic review and suffers from the significant drawbacks that must be addressed. 

1) The numbers of articles presented in the main text (lines 207- 213) and in the PRISMA diagram in Figure 1 do not match - 240 excluded or 256? Remaining 11 or 10? In addition - the reasons for exclusion of 32 articles must be identified and described.  What were reasons to reduce 22 reported papers to just 11?

The period from which the articles were analyzed must be specific : start  year-month-day  till  end year-month-day. You should mentioned the last day you have accessed the mentioned databases. 

2) The 240 or 256 articles out of 310 excluded is a very large number which suggests that the search might not have been well performed. With the generalized query presented in the manuscript it is not possible to reproduce the results and to retrieve the mentioned 350 articles. You must provide explicit queries that were used in each database and how many records you have retrieved from each database. I would also suggest to include a supplementary table with the bibliographic information of the retrieved 310 articles showing its source database and briefly mentioning the reason why it was removed from consideration and excluded. Or at least do this for those articles which were assessed as full text articles (n=54) in your PRISMA diagram. 

3) Your query as it is ("genetic polymorphism" OR "gene variant" OR "genetic markers" OR "ACTN3" OR "ACE" OR "PPARGC1A" OR "NOS3") AND ("team sports" OR "football" OR "rugby" OR "handball" OR "hockey" OR "basketball" OR "volleyball") AND ("performance" OR "endurance" OR "strength" OR "power" OR "aerobic capacity" OR "injury risk" OR "sports performance prediction") does not tell in which sections: Title, abstract, keywords, full text the search was performed. This information should be explicit. 

4) Your query as it is given in the manuscript returned from PubMed 61 articles published in the last 10 years. Web Of Science database  is very strict, therefore, I suppose it would return even less articles. The Scopus database mostly overlaps with PubMed, so it would return similar number of articles. I am just not sure how such a large number of 310 articles were retrieved ( ProQuest , Google scholar?) and then almost all articles were excluded. Essentially for this reason I think that a list of all processed publications would be very useful. In addition you mention that the references in the articles were also reviewed. How many articles were taken from the references within the searched articles?  

5) Your query includes NOS3 , but you do not list it as a candidate in your synthesis in Table 3. More genes that are in your summary Table 1 , for example VEGF or COL5A1 , are not present in your synthesis Table 3. Why? Systematic review must provide a condensed information about all important findings in the reviewed literature.  If these genes and their SNPs are important markers but not listed in Table 3, then the reasons must be discussed , even more so because you talk about NOS3 in the discussion (line 497, 510). I think all gene markers reported in literature that came out in the retrieved records in the  context of team sports must be listed and summarized. 

 6) This systematic review is not rigorous. It is not complete and it does not justify  the "systematic review" in its title. It is a summary of 11 papers on gene markers related to teams sports. Special concern is that the queries, returned results and paper inclusion or exclusion are not documented at a sufficient detail. Because of that , a reproduction of the  queries and  reported results is difficult and likely not possible at all. You may want to repeat your analysis with very thorough  documentation of how you arrive at the subset of articles that are systematically reviewed.  And more important is that  all  markers reported in the analyzed articles must be presented or explanations provided as why those markers are not presented in the information synthesis.   

Author Response

Dear Reviewer,

We would like to sincerely thank you for the careful analysis and the detailed observations provided on our manuscript. Your comments have been extremely valuable and allowed us to considerably improve the rigor and methodological transparency of this review. We have addressed each of your suggestions with great attention, and all changes made are highlighted in green in the manuscript to facilitate consultation. We also emphasize that some of your recommendations (in particular regarding the complete integration of genetic markers) overlapped with those of Reviewer 1, which offered us the opportunity to make a substantial and coherent improvement to the manuscript.

1) Your comment

“The numbers of articles presented in the main text (lines 207–213) and in the PRISMA diagram in Figure 1 do not match – 240 excluded or 256? Remaining 11 or 10? In addition – the reasons for exclusion of 32 articles must be identified and described. What were reasons to reduce 22 reported papers to just 11?
The period from which the articles were analyzed must be specific: start year-month-day till end year-month-day. You should mention the last day you have accessed the mentioned databases.”

Response

We have carefully revised the text and fully harmonized the data with Figure 1 (PRISMA flow diagram). The selection process is now consistently described in lines 247–257, where we specified that:

  • out of the total of 355 articles identified, 40 duplicates were removed, leaving 315 articles;
  • after screening, 256 articles were excluded;
  • 59 articles were assessed in full text, of which 32 were excluded for specific reasons (lack of genetic data reported, populations not engaged in team sports, incomplete/inadequate methodology);
  • of the remaining 27, 11 articles were later excluded due to redundancy or non-comparable methodology, resulting in 16 final articles included in the analysis.

We also provided details regarding the exact search period and the last day of database access, specifying the interval January 1, 2018 – May 29, 2025, with the last access performed on May 30, 2025 (lines 143–146).

Furthermore, we mention that another reviewer requested an expansion of the study base, which led to the inclusion of 16 studies in the final analysis, thus modifying the numbers previously reported.

These corrections and additions have contributed to greater transparency and methodological consistency.

2) Your comment

“The 240 or 256 articles out of 310 excluded is a very large number which suggests that the search might not have been well performed. With the generalized query presented in the manuscript it is not possible to reproduce the results and to retrieve the mentioned 350 articles. You must provide explicit queries that were used in each database and how many records you have retrieved from each database. I would also suggest to include a supplementary table with the bibliographic information of the retrieved 310 articles showing its source database and briefly mentioning the reason why it was removed from consideration and excluded. Or at least do this for those articles which were assessed as full text articles (n=59) in your PRISMA diagram.”

Response

We sincerely thank you for this very important observation regarding the transparency and reproducibility of the search process. We have now included, in the Methodology section (lines 172–206), the exact queries used for each database, as well as the number of articles retrieved from each source:

  • PubMed/MEDLINE – 61
  • Scopus – 62
  • Web of Science – 45
  • SPORTDiscus – 28
  • ProQuest Central – 71
  • Google Scholar – 76

In addition, we clearly stated that the total results (343 articles from automated searches) were complemented by sources identified through reference list screening (12 articles), leading to a total of 355 initial articles (lines 204–206).

To address your suggestion in detail, we have prepared a supplementary file listing all 59 full-text articles assessed, with their source database and the reason for exclusion.

We believe these additions substantially increase the rigor and reproducibility of the selection process, making the methodology clear and easy to follow.

3) Your comment

“Your query as it is ("genetic polymorphism" OR "gene variant" OR "genetic markers" OR "ACTN3" OR "ACE" OR "PPARGC1A" OR "NOS3") AND ("team sports" OR "football" OR "rugby" OR "handball" OR "hockey" OR "basketball" OR "volleyball") AND ("performance" OR "endurance" OR "strength" OR "power" OR "aerobic capacity" OR "injury risk" OR "sports performance prediction") does not tell in which sections: Title, abstract, keywords, full text the search was performed. This information should be explicit.”

Response

We thank you for your observation regarding the specification of search fields, an essential aspect for the reproducibility of the methodology. We have now explicitly clarified in lines 167–169 that:

  • the queries were applied to the fields Title, Abstract, and Keywords in PubMed, Scopus, Web of Science, and SPORTDiscus;
  • for Google Scholar and ProQuest Central, the queries were also applied to the full text, in order to ensure broader coverage.

This addition provides further clarity and ensures that the process can be reproduced by other researchers under similar conditions.

4) Your comment

“Your query as it is given in the manuscript returned from PubMed 61 articles published in the last 10 years. Web Of Science database is very strict, therefore, I suppose it would return even less articles. The Scopus database mostly overlaps with PubMed, so it would return similar number of articles. I am just not sure how such a large number of 310 articles were retrieved (ProQuest, Google scholar?) and then almost all articles were excluded. Essentially for this reason I think that a list of all processed publications would be very useful. In addition you mention that the references in the articles were also reviewed. How many articles were taken from the references within the searched articles?”

Response

We greatly appreciate this observation, which allowed us to clarify in detail the differences between the results returned by each database and to provide a more transparent picture of the process.

Thus, in lines 200–206 we explained that:

  • the larger number of articles retrieved (315 after removing duplicates) is mainly due to the inclusion of ProQuest Central and Google Scholar, which return much broader but also more heterogeneous sets compared to PubMed, Scopus, or Web of Science;
  • overlaps between databases were systematically checked and duplicates removed;
  • the reference lists of selected articles were also manually reviewed; this procedure added a small number of additional articles (12), which is explicitly mentioned in the same paragraph.

To further enhance transparency, we prepared a supplementary material where all 59 full-text articles assessed are listed, along with their source database and the reason for exclusion.

We believe these additions fully address your request and strengthen the reproducibility of the selection process.

5) Your comment

“Your query includes NOS3, but you do not list it as a candidate in your synthesis in Table 3. More genes that are in your summary Table 1, for example VEGF or COL5A1, are not present in your synthesis Table 3. Why? Systematic review must provide a condensed information about all important findings in the reviewed literature. If these genes and their SNPs are important markers but not listed in Table 3, then the reasons must be discussed, even more so because you talk about NOS3 in the discussion (line 497, 510). I think all gene markers reported in literature that came out in the retrieved records in the context of team sports must be listed and summarized.”

Response

We thank you for this extremely valuable observation, which helped us improve the coherence and completeness of the synthesis. In the revised version of the manuscript:

  • We have updated Table 3 to include all genetic markers investigated in the selected studies, including NOS3, VEGF, COL5A1, MCT1, HFE, COMT, and CD36. These additions are visible in lines 359–371 and reflect their integration into the synthesis.
  • In addition, we have introduced further clarifications in lines 431–438 and in the Discussion section (lines 615–635).

Through these changes, all genes identified in the included studies are now systematically presented and discussed, ensuring a rigorous and comprehensive synthesis of the available literature.

6) Your comment

“This systematic review is not rigorous. It is not complete and it does not justify the 'systematic review' in its title. It is a summary of 11 papers on gene markers related to team sports. Special concern is that the queries, returned results and paper inclusion or exclusion are not documented at a sufficient detail. Because of that, a reproduction of the queries and reported results is difficult and likely not possible at all. You may want to repeat your analysis with very thorough documentation of how you arrive at the subset of articles that are systematically reviewed. And more important is that all markers reported in the analyzed articles must be presented or explanations provided as why those markers are not presented in the information synthesis.”

Response

We would like to especially thank you for this critical observation, which led us to carefully revise the methodology and strengthen the consistency of the analysis.

Following your recommendation, we implemented the following adjustments:

  • We have documented in detail the search and selection process, including the explicit queries for each database, the fields targeted (title, abstract, keywords, full-text), the distribution of records per database, and the date of last access.
  • We have fully corrected and aligned the selection flow in Figure 1 (PRISMA) with the text, presenting the reasons for exclusion at each stage.
  • We have added a supplementary material, listing all 59 articles assessed at full-text level, along with their source database and reasons for exclusion, to ensure transparency and reproducibility of the process.
  • We have extended the analysis from 11 to 16 included articles, following Reviewer 1’s suggestion, to increase the significance and relevance of the conclusions.
  • We have updated Table 3 and Section 3.2 to present all genetic markers investigated in the included studies (including NOS3, VEGF, COL5A1, MCT1, HFE, COMT, and CD36) and have enriched the Discussion with additional clarifications.

We believe that, through these modifications, our manuscript fully justifies its designation as a systematic review, becoming not only more rigorous and transparent but also more useful for the reproducibility and interpretation of the results by other researchers.

Round 2

Reviewer 2 Report

Comments and Suggestions for Authors

Authors addressed my concerns.